# Identification and Characterization of *Clostridium perfringens* Atypical CPB2 Toxin in Cell Cultures and Field Samples Using Monoclonal Antibodies

**DOI:** 10.3390/toxins14110796

**Published:** 2022-11-17

**Authors:** Anna Serroni, Claudia Colabella, Deborah Cruciani, Marcella Ciullo, Silvia Crotti, Paola Papa, Antonella Di Paolo, Marco Gobbi, Katia Forti, Martina Pellegrini, Romolo Salini, Nicoletta D’Avino, Monica Cagiola, Giovanni Pezzotti, Antonio De Giuseppe

**Affiliations:** 1Istituto Zooprofilattico Sperimentale dell’Abruzzo e del Molise “G. Caporale” (IZSAM), Campo Boario, 64100 Teramo, Italy; 2Istituto Zooprofilattico Sperimentale dell’Umbria e delle Marche “Togo Rosati” (IZSUM), Via G. Salvemini 1, 06126 Perugia, Italy

**Keywords:** *C. perfringens*, atypical CPB2 toxin, sELISA

## Abstract

A direct sandwich enzyme-linked immunosorbent assay (sELISA) was developed for the detection of the atypical β2-toxin (CPB2) of *Clostridium perfringens*. Polyclonal (PAbs) and monoclonal (MAbs) antibodies were previously obtained employing recombinant CPB2 produced in the baculovirus system as antigen. In the current study, PAbs were used as capture molecules, while purified MAbs conjugated to horseradish peroxidase (MAbs-HRP) were used for the detection of atypical CPB2 toxin. MAbs 5C11E6 and 2G3G6 showed high reactivity, sensitivity and specificity when tested on 232 *C. perfringens* cell culture isolates. In addition, a reactivity variation among different strains producing atypical CPB2 toxin was observed using the conformation-dependent MAb 23E6E6, suggesting the hypothesis of high instability and/or the existence of different three-dimensional structures of this toxin. Results obtained by sELISA and Western blotting performed on experimentally CPB2-contaminated feces revealed a time-dependent proteolytic degradation as previously observed with the consensus allelic form of CPB2. Finally, the sELISA and an end-point PCR, specific for the atypical *cpb2* gene, were used to test field samples (feces, rectal swabs and intestinal contents) from different dead animal species with suspected or confirmed clostridiosis. The comparison of sELISA data with those obtained with end-point PCR suggests this method as a promising tool for the detection of atypical CPB2 toxin.

## 1. Introduction

*Clostridium perfringens* is a commensal gram-positive bacterium that is found in gastrointestinal tract of humans and other animals, and it is characterized by worldwide distribution [1]. This sporogenic anaerobic bacterium can cause a wide range of diseases from enteritis and enterotoxaemia in domestic and wild animals to serious forms of food poisoning in humans [2,3]. The pathogenic mechanism of *C. perfringens* is explicated through the production of over 17 different extracellular toxins. The newly updated toxinotyping scheme classified *C. perfringens* into seven toxinotypes (A–G) based on its ability to produce the six major enterotoxins, such as the alpha (CPA), beta (CPB), epsilon (ETX), iota (ITX), enterotoxin (CPE) and the necrotic enteritis B-like toxins (NetB) [4,5]. In the last decade, different toxins, defined as minor, have been identified. Among these, atypical CPB2 toxin—first isolated from a piglet with necrotic enteritis—has received significant attention. This toxin shared with the CPB toxin a similar pathogenic mechanism; this occurs even if they have low genetic homology [6].

The CPB2 toxin is a 28 kDa protein encoded by the *cpb2* gene located on a large plasmid in two allelic forms: the consensus (*cons*) CPB2, isolated mainly from non-porcine animals [2,6,7,8], and the less toxic atypical (*aty*) CPB2 [3,9]. Among *C. perfringens* isolates from non-porcine species, the atypical CPB2 toxin displays 62.3% identity and 80.4% similarity to that encoded by the *cons cpb2* allele and, surprisingly, it is not always expressed [3]. However, both allelic variants of *cpb2*, particularly the atypical variant, occur frequently in animals without signs of acute enteric disease [10]. The pathogenic role of CPB2 toxin is not clear, although Benz et al. recently established its ability to form highly cation-selective channels in lipid bilayers [11]; other authors reported that CPB2 induced apoptosis and inflammatory response in intestinal porcine epithelial (JPEC-J2) cells [12,13]. Since *C. perfringens* is a commensal associated with the microbiota of healthy animals, the diagnosis of enterotoxaemia and gas gangrene should be based essentially on the detection of toxins in the intestinal contents. Nowadays, the diagnostic process involves the isolation of *C. perfringens* in selective media combined with biochemical characterization and the identification of toxin genes by PCR and qPCR [14].

The toxinotyping of the isolated strains is often time-consuming and related to in vitro growth and biomolecular typing assays; moreover, the presence of the *cpb2* gene, especially the atypical variant, does not always correspond to the production of the toxin itself [3]. Therefore, evidence of different toxins in pathological material is critical for proper identification of the pathology and typing of its causative agent and thus, the identification of the *cpb2* positive (+) strains by the PCR method may not be sufficient to link the disease with the presence of the toxin. To assess the role of the CPB2 toxin during enterotoxaemia, it is necessary to use techniques that are able to identify CPB2 in clostridiosis, such as immunoenzymatic assay (ELISA).

Recent findings report the development of an ELISA to detect consensus CPB2 toxin in the neonatal pig intestine, and this method has also highlighted the proteolytic degradation of the wild-type consensus CPB2 [3,6,15,16,17,18,19]. Other authors have generated MAbs using consensus CPB2-derived polypeptides, but the use of these molecules was limited exclusively to the evaluation of the in vitro biological activity of purified recombinant consensus CPB2 [13,20].

The present work focused on the use of an immunoenzymatic method (sELISA) for the identification and characterization of *C. perfringens* atypical CPB2 toxin in both cell culture supernatants and field samples of different animal species with suspected or confirmed clostridiosis. The results were successfully achieved by using recently developed PAbs and MAbs [21].

## 2. Results

### 2.1. Evaluation of MAbs Capture Abilities

Twenty hybridomas secreting anti-atypical CPB2 MAbs were recently obtained and tested by immunological assay [21]. The capture abilities of the purified MAbs were assessed by sELISA using *C. perfringens* atypical *cpb2*+ strain culture supernatant. Out of a total of 20 different hybridomas, only seven MAbs were able to capture atypical CPB2 toxin when adsorbed on the plate. Five of these MAbs displayed a mild capture ability (5B4G9G11, 5B4F11, 4E10E11, 4E10E10 and 2G3D10, data not shown), whereas MAbs 5C11E6 and 2G3G6 showed a very high binding ability. Figure 1 shows the results obtained in sELISA using MAb 2G3G6 as catcher and MAb 5C11E6-HRP as the detection system. Comparable results were obtained using MAb 5C11E6 as capture molecule and MAb 2G3G6-HRP as detector (Figure 1).

### 2.2. Identification of C. perfringens cpb2+ Strains and Determination of the Specificity and Sensitivity of MAbs

Toxinotyping of *C. perfringens* strains was performed by multiplex-PCR, whereas a simplex PCR was used to detect the atypical allelic forms of the *cpb2* gene. Out of 232 strains analyzed, 105 were identified as atypical *cpb2*+, 27 as consensus *cpb2*+ and the remaining 100 were *cpb2* negative (−). All supernatants of these strains were used to determine the specificity and sensitivity of the anti-CPB2 MAbs. Initially, these studies were focused on evaluating the performance of fifteen anti-CPB2 MAbs conjugated to peroxidase as a detection system using *C. perfringens* culture supernatants. Specifically, 36 atypical *cpb2*+ strains, 5 consensus *cpb2*+ and 2 *cpb2*− strains were selected and analyzed by sELISA using the whole MAbs panel. The results obtained for these strains showed high OD_450_ values for the atypical *cpb2*+ strains and significantly lower OD_450_ values for consensus *cpb2*+ and *cpb2*− strains, although a variability among all tested MAbs was observed for atypical *cpb2*+ strains (data not shown). The remaining culture supernatants (69 atypical *cpb2*+, 22 consensus *cpb2*+ and 98 *cpb2*− *C. perfringens* strains) were subjected to sELISA using only MAbs 5C11E6, 2G3G6, 4E10E11 directed against linear epitopes and conformation-dependent MAb 23E6E6, which have been previously characterized [21].

Overall, these four MAbs recognized the atypical CPB2 toxin produced by almost all *C. perfringens* strains genotyped as atypical *cpb2*+, considering the high OD_450_ values observed. Specifically, among 105 *C. perfringens* atypical *cpb2*+ strains, 79 samples displayed very high and similar OD_450_ values (data not shown). In another 19 samples out of a total of 105, the atypical CPB2 protein was evidenced by variable OD_450_ values among the selected MAbs (Table 1).

This variability reflected a lower sensitivity for MAb 4E10E11 than for MAbs 5C11E6 and 2G3G6 and the best reactivity for MAb 5C11E6. The remaining 7 atypical strains out of 105 displayed low OD_450_ values (Appendix A). The high specificity of these MAbs was also confirmed by sELISA analysis with strain culture supernatants genotyped as consensus *cpb2*+. Out of a total of 27 typical strains analyzed, only one sample (C28) showed higher OD_450_ values for all four MAbs, while those of the remaining strains ranged between 0.1 and 0.13 (Appendix A).

The *cpb2*− strains analyzed by sELISA showed higher OD_450_ values in only one sample (C66) and a very slight background for MAb 4E10E11 in 9 out of 100 samples but with extremely low OD_450_ values (0.1–0.134), followed by MAb 5C11E6 (samples C64 and C65) (Appendix A).

### 2.3. Degradation of Atypical CPB2 Toxin in Experimentally Contaminated Faeces

Previous studies showed that a major limitation in the development of an ELISA was represented by to the proteolytic degradation of the consensus CPB2 toxin in field samples [15]. The degradation of atypical CPB2 was determined in both sELISA and Western blotting. According to the results, a consistent and time-dependent degradation of CPB2 atypical allelic form was observed (Figure 2).

Initially, degradation of CPB2 was determined in Western blotting using hyperimmune serum (Figure 3a).

As shown in Figure 3a, the absence of the full-size protein and the presence of a degraded polypeptide of approximately 10–12 kDa after 24 h of incubation of the contaminated feces at room temperature were observed. After a longer incubation period, this fragment was also degraded (data not shown). These results were confirmed using all anti-CPB2 MAbs. In particular, MAbs 2G3G6, 3G4C8 and 2G3D10 were able to detect the degraded product of 10–12 kDa (Figure 3b,d,e) as was MAb 5B4G9G1, to a lesser extent (Figure 3c).

### 2.4. Atypical CPB2 Toxin Detection on Field Samples and PCR Comparison

Feces, rectal swabs and intestinal contents were recovered from dead animals with suspected or confirmed clostridiosis. These samples were subjected to PCR for atypical *cpb2* gene and sELISA.

All the matrices from 30 animals (samples 1–30) out of 74, with the exception of unavailable data for some matrices (samples number 5, 14, 17, 24 for feces, 5 and 24 for the rectal swab), were PCR negative (Appendix A).

For almost all these matrices, the immunoenzymatic assay showed low OD_450_ values. Particularly, all the values were below 0.1 with the exception of three samples of feces (26, 27 and 29) and intestinal contents (13, 28 and 30) and four rectal swabs (25, 27, 29 and 30) (Appendix A). In another 44 animals, the atypical *cpb2*+ gene was detected by PCR in all the intestinal content samples, with the exception of sample 51 and samples 73 and 74, whose results appeared doubtful (Table 2).

In particular, in 12 animals the PCR was positive for all three matrices (samples 31–42) and for the two available matrices in six further samples (feces for samples 43–46 and 47–48 were unavailable for rectal swab). Furthermore, in the absence of two matrices for samples 49 and 50, the presence of *C. perfringens* could only be detected on feces and intestinal contents, respectively. In the remaining 24 animals (51–74, with the exception of samples 71–74), the biomolecular test detected the atypical *cpb2*+ gene in the feces of only five other animals (samples 51, 52 and 54–56). Similarly, in the rectal swab, the atypical *cpb2*+ gene was detected in only two samples (51 and 53). Overall, these results clearly showed that positive PCR outcomes were mainly found in the intestinal content matrix.

Similar to PCR, the best results in sELISA were obtained on intestinal contents, where OD_450_ values were well above 0.1. In this case, only OD_450_ from samples 38, 41 and 57 showed values similar to the background found in animals where the PCR did not detect the atypical *cpb2*+ gene (Appendix A and Table 2). In the other two matrices, the results obtained in sELISA were less expected, as several PCR-positive samples showed OD_450_ values similar to the PCR-negative samples (Appendix A).

Statistical analysis of sELISA data showed good variability for each type of sample. As expected, the best match between the sELISA and PCR data was obtained using the intestinal content as sample, showing higher OD_450_ values (Figure 4).

The result obtained in the Friedman test was highly significant and showed a *p* value < 0.0001, highlighting relevant differences between the compared matrices. Furthermore, pairwise comparisons were considered, showing significant results between the matrices: a *p* value = 0.043 was obtained comparing intestinal contents vs. feces; a *p* value < 0.0001 was obtained comparing intestinal contents vs. rectal swabs; the comparison between rectal swabs and feces showed a *p* value of 0.061 (Table 3).

## 3. Discussion

The pathogenicity of *Clostridium perfringens* is attributed to the production of different major and minor toxins such as CPB2. To date, the role of this toxin during enterotoxaemia, in particular the atypical form, is still not clear. The study of the biological activity of this toxin is limited by the lack of reagents for the development of an immunoenzymatic assay. Recent findings showed that one of the major limitations in the standardization of an ELISA method for the detection of CPB2 during clostridiosis was related to the degradation of the toxin itself in the samples [9]. Another drawback was related to the limited availability of specific MAbs for CPB2 and especially for the atypical form. In fact, some authors reported the use of polyclonal and monoclonal antibodies directed exclusively against the consensus CPB2, which showed low reactivity towards the CPB2 atypical form [15,22]. Other authors described the use of MAbs generated by consensus CPB2-derived polypeptides, limited to the study of the biological activity and the pathogenic mechanisms of consensus CPB2 [13,20]. Recently, a panel of MAbs directed against atypical CPB2 was developed and tested in sELISA [21]. These reagents showed different reactivity variations in both immunoassay and Western Blotting.

In preliminary studies, fifteen MAbs were tested, and the sELISA was subsequently performed using only MAbs 5C11E6, 2G3G6, 4E10E11 and the conformation-dependent MAb 23E6E6. The first two MAbs were chosen because of their excellent performance in terms of specificity and sensitivity, as well as their high capture abilities (Figure 1), while MAb 4E10E11 was randomly selected from those displaying medium binding ability (data not shown).

These MAbs were unable to detect CPB2 in seven samples genotyped as atypical *cpb2*+, with the exception of sample C20, which was considered doubtful (Appendix A). Interestingly, Jost and colleagues reported that several *C. perfringens* isolates from non-porcine species carrying the *cpb2* atypical gene were not able to express CPB2 toxin due to a frameshift mutation that resulted in the synthesis of a truncated atypical CPB2 of 4.5 kDa [3]. Furthermore, the same authors observed that in other *C. perfringens* strain culture media, the low level or even the absence of RNA transcription of the atypical CPB2 toxin resulted in defective expression of the toxin itself [3]. In fact, the absence of CPB2 in seven culture supernatants of atypical *cpb2*+ strains could be due to a frameshift mutation or, alternatively, to the lack of gene expression at the RNA transcription level.

As reported in Table 1, some samples (i.e., C4, C5, C18 and C19) displayed CPB2 expression levels with remarkable variability, thus suggesting that other mechanisms could affect the level of gene expression at least for strains growing in culture medium [3]. However, these MAbs displayed no cross-reactivity with other strains producing consensus CPB2 showing high specificity (Appendix A). These results showed that anti-atypical CPB2 MAbs were probably directed against the highest amino acid variability regions, considering the comparison between the two allelic forms of CPB2 [3]. Conversely, since all MAbs detected the toxin in most strain culture supernatants and atypical CPB2 of non-porcine origin showed rather frequent amino-acid variations, it can be assumed that they recognized and conserved different epitopes among the various atypical CPB2 produced by these strains.

In this study, conformation-dependent MAb 23E6E6 was also employed in order to assess whether atypical CPB2 could reflect different conformational structures among different strains, considering the polymorphism of the atypical *cpb2* gene [16,23]. In sELISA, this MAb showed the highest degree of variability in reactivity in addition to its lower sensitivity compared to the other MAbs (Table 1). These results suggest that the loss of reactivity could be related to the conformational instability of native atypical CPB2, as previously reported [6,20], without excluding the existence of different three-dimensional structures of the atypical CPB2 toxin.

The pathogenic role of CPB2 is not clear and therefore may act in synergy with other major toxins, such as alpha-toxin, thereby increasing its own toxic effect [1,20,24]; moreover, this aspect, as well as the regulatory mechanisms involved in CPB2 expression, could influence its association as a cause of necrotic enteritis [16,22]. In this context, and based on the results obtained with MAb 23E6E6-HRP, it could be assumed that the toxicity of CPB2 and/or its synergy action with CPA toxin could be linked to different structural conformations due to amino acid sequence variations or to its lower conformation stability.

In order to evaluate the sELISA method for the detection of CPB2 in field samples, preliminary tests were performed using artificially contaminated feces. The results obtained in sELISA confirmed the existence of a time-dependent proteolytic degradation of atypical CPB2 on experimentally contaminated feces [15,22]. However, high OD_450_ values were observed in this new sELISA even after a 24-h incubation period. Degradation of atypical CPB2 in contaminated feces was also observed in Western blotting, indicating the strong performance of this sELISA in detecting the 10–12 kDa polypeptide fragment.

Based on these results, field samples (feces, intestinal contents and rectal swabs) were simultaneously analyzed by sELISA and PCR. Although the PCR method does not represent the gold standard, it was nevertheless able to detect the atypical *cpb2* gene mainly in the intestinal contents. For this matrix, in 40 out of a total of 74 dead animals with suspected or confirmed clostridiosis belonging to the bovine, ovine and caprine, the presence of the *C. perfringens* gene was detected with a percentage of 54.8%. In 20 out of 63 feces samples (with the exception of unavailable data), the *cpb2* gene was detected with a percentage of 31.7%, followed by rectal swabs where only 18 samples out of 68 tested positive at a percentage of 26.5%. In the immunoenzymatic method, the highest OD_450_ values were also observed for the intestinal contents. These results were supported by statistical analysis (Figure 4 and Table 3) and therefore the intestinal content matrix was chosen to evaluate the performance of both methods. However, based on these results, two observations can be made: (i) 30 out of 74 samples were PCR negative (including feces and rectal swabs) and for most of them, sELISA OD_450_ values were extremely low and below 0.1 (Appendix A); (ii) for several matrices, all PCR-positive samples displayed OD_450_ values ≥ 0.1.

Based on these considerations, and despite the low statistical significance of the number of intestinal matrices analyzed, a potential cut-off of 0.129 was estimated for the immunoassay, taking into account all OD_450_ values of the intestinal contents of the 30 animals that tested negative for PCR (Appendix A). Therefore, all intestinal contents that showed OD_450_ values ≥ 0.129 in sELISA were considered positive. Notably, it was not possible to establish the elapsed time between the death of the animal and the processing of the sample, considering that CPB2 degradation is time-dependent and that the level of expression may vary depending on the strain of *C. perfringens* and/or the matrix in which it is present [3]. For these reasons, the possibility that for some samples (i.e., 38, 41, 46, 57 and 68) the low OD_450_ values could be associated with and/or dependent on these issues (Table 2) cannot be ruled out. Conversely, sample 30 tested positive in the sELISA against a negative PCR outcome, like sample 28, although bordering with the estimated cut-off of 0.129. In this case, sELISA could be considered a false positive result or a PCR aberration probably related to nucleotide variability of the *cpb2* gene [3].

Considering these results, it is difficult to determine which methods have the best performance in terms of specificity and sensitivity. The aim of this work was not focused on the standardization of an immunoenzymatic assay but on the evaluation of the functionality of monoclonal and polyclonal antibodies and the ability of these reagents (particularly MAb 5C11E6) to detect the atypical CPB2 toxin on matrices derived from livestock animals (bovine, ovine and caprine). To date, there is no gold standard method, nor can the PCR used in this study be considered a reference since the nucleotide sequence of the *cpb2* gene can show some variability. On the other hand, the rapid degradation of the atypical CPB2 in field samples could represent a limitation in terms of sensitivity. However, an improvement of the sELISA could be provided by reducing the time between sample collections and processing in order to ensure the integrity of the toxin as well as by using Western blotting, which can detect atypical CPB2 degradation fragments. Furthermore, this specific immunoenzymatic assay for atypical CPB2 was performed for the first time and on a preliminary basis on bovine, ovine and caprine. Future studies involving this sELISA on a larger and statistically significant sample panel, compared with further biomolecular techniques, could undoubtedly lead to a standardization of this method in order to perform a qualitative as well as a quantitative test.

Furthermore, the use of sELISA could provide important information on the link between the degree of disease manifestation and the presence of atypical CPB2 in fresh pathological material during clostridiosis. Finally, the use of the reagents in the research field could also make an important contribution to the understanding of the pathogenesis and biological activity of atypical CPB2 in both veterinary and human medicine.

## 4. Conclusions

In this study, the detection of the atypical CPB2 toxin of *C. perfringens* was investigated using recently developed PAbs and MAbs. The results obtained from bacterial cell culture, artificially CPB2-contaminated feces and field samples suggested that these reagents could be used for the setting up of an enzyme immunoassay. Moreover, the use of sELISA and Western blotting confirmed the existence of a time-dependent proteolytic degradation of the atypical CPB2 toxin. Additional studies are required for the development and standardization of this method, which could be used to understand the pathogenic mechanisms of atypical CPB2 toxin.

## 5. Materials and Methods

### 5.1. Production and Purification of Monoclonal Antibodies (MAbs)

Polyclonal and monoclonal antibodies were obtained as previously described [21]. Specifically, twenty hybridomas were grown in 200 mL of Dulbecco’s Modified Eagle Medium (DMEM) containing 10% fetal bovine serum (Merck, Rahway, NJ, USA) in Nunclon Delta Surface cellular culture flasks (Thermo Fisher Scientific, Waltham, MA, USA) at 37 °C and 5% CO_2_. After 12 days, the supernatants were collected, centrifuged at 400× *g* for 15 min, filtered and purified. Affinity chromatography purification was performed using the “rProtein A GraviTrap™ kit” (GE Healthcare Life Sciences, Chicago, IL, USA) following the manufacturer’s instructions. The MAbs purity level was verified by SDS-PAGE gels stained with Coomassie Brilliant Blue (BIO-RAD), and their quantification was performed using Bradford assay. The purified MAbs were conjugated with horseradish peroxidase (HRP) using the “Lightning-Link^®^ HRP Conjugation Kit” (INNOVA Biosciences, Montluçon, France) according to the manufacturer’s instructions.

### 5.2. Genotyping and Molecular Characterization

Two hundred and thirty-two strains of *C. perfringens* from bovine, ovine, caprine and other animals were collected and stored in the laboratory collection (IZSUM). For isolation, *C. perfringens* strains were grown anaerobically on blood agar plates containing 5% defibrinated sheep’s blood at 37 °C for 48 h. Suspected colonies were identified by RAPID ID 32A (Biomérieux^®^, Lyon, France). *C. perfringens* strains were then grown anaerobically on TPGY medium (2% yeast extract, 0.1% glucose, 3% tryptone, 0.5% meat peptone and 0.1% L-cysteine) at 37 °C for 48 h. Genomic DNA of collected *C. perfringens* strains was extracted using the QIAamp Fast DNA Stool Mini Kit (QIAGEN^®^, Hilden, Germany) following the manufacturer’s instructions. To determine the toxinotype of each isolate, the extracted DNA was subjected to a multiplex-PCR [5]. The atypical form of the *cpb2* gene was evidenced by end-point PCR as previously described [21].

Seventy-four field samples (feces, rectal swabs and intestinal contents) derived from bovine, ovine and caprine with suspected or confirmed clostridiosis were sampled and diluted 1:1 in phosphate-buffered saline (PBS) containing protease inhibitors to prevent atypical CPB2 toxin degradation. To establish the presence of the atypical *cpb2* gene, 200 µL of diluted samples were subjected to DNA extraction, and the specific target was amplified by end-point PCR as reported above. Finally, all the collected samples were tested with a homemade ELISA (sELISA) to reveal the presence of atypical CPB2 toxin.

### 5.3. Artificially Contaminated Faeces

Culture supernatant of *C. perfringens* containing atypical CPB2 was added in a 1:5 ratio to the bovine feces. Contaminated samples were incubated at room temperature (RT). Aliquots were collected at 0, 4 h, 8 h, 24 h, 48 h and 72 h after contamination and frozen at −80 °C until use. After thawing, the contaminated feces were first subjected to sELISA using rabbit hyperimmune serum as the catcher and MAbs 5C11E6-HRP and 2G3G6-HRP as the detection system. Subsequently, the same samples were analyzed by Western blotting using both polyclonal antibodies and the whole MAbs panel [21].

### 5.4. ELISA Sandwich (sELISA)

Ninety-six-well MaxiSorp microplates (MaxiSorp Nunc International, Denmark) were coated with rabbit polyclonal antibody diluted 1:10,000 in carbonate/bicarbonate buffer (pH 9.6) at +4 °C over night (ON). After three washes with PBST (phosphate-buffered saline with 0.05% Tween 20), *C. perfringens* culture media (diluted 1:5 in PBST with 1% yeast extract) were added and incubated for 1 h at 37 °C. The plates were further washed three times, and subsequently, each MAb-HRP was added at the appropriate dilution in PBST. After 1 h of incubation at 37 °C, the plates were washed and the TMB chromogenic substrate was added and incubated for 15 min at RT. The reaction was stopped with 0.5 M H_2_SO_4_ and the optical density at 450 nm (OD_450_) was measured.

The same procedure was used for artificially contaminated feces (diluted 1:5 in PBST with 1% yeast extract) and field samples (two-fold dilution in PBST with 1% yeast extract). MAbs 5C11E6-HRP (diluted 1:20,000) and 2G3G6-HRP (diluted 1:10,000) were used as detection systems for artificially contaminated feces, while only MAb 5C11E6-HRP was used to detect the presence of atypical CPB2 in field samples.

Similarly, the capture abilities of the MAbs 5C11E6 and 2G3G6 (diluted 1:5000) were tested using atypical wild-type CPB2 (two-fold dilutions in PBST with 1% yeast extract) with the same MAbs-HRP used as the detection system.

### 5.5. Statistical Analysis

Statistical analyses were performed using R Core Team software [25]. A first analysis was carried out to assess the performance of the sELISA for each sample type in comparison to the PCR. In addition, a non-parametric test (Friedman test for k dependent samples) was applied to check the differences in the sELISA among the three different field samples (feces, rectal swabs and intestinal contents) and a post-hoc test was also performed to verify the difference between all possible pairs (Multiple pairwise comparisons using Nemenyi’s procedure/Two-tailed test) [26].

### 5.6. Western Blotting

Contaminated feces with *C. perfringens* culture supernatant containing atypical CPB2 were mixed with 4xNu-PAGE sample buffer (Thermo Fisher Scientific) containing 10 mM DTT and denatured at 99 °C for 5 min. The samples were resolved by SDS-PAGE in 12% acrylamide pre-cast NuPAGE gels and transferred onto a PVDF membrane (Thermo Fisher Scientific). After 2 h incubation at RT with a TBST blocking solution (20 mM Tris–HCl, 150 mM NaCl, 0.05% Tween 20 (*v*/*v*) at pH 7.4) containing 5% dry milk, the membranes were incubated ON with both hybridoma cultures containing anti-CPB2 MAbs and hyperimmune serum, respectively. The membranes were washed three times and incubated for 1 h with anti-mouse (1:20,000, Thermo Fisher Scientific) and anti-rabbit (1:25,000, Thermo Fisher Scientific) HRP-conjugated MAb for anti-CPB2 MAbs and polyclonal antibodies, respectively. The immune reactions were visualized by chemiluminescence using the Super Signal West Pico Substrate kit (Thermo Fischer Scientific).

## Figures and Tables

**Figure 1 toxins-14-00796-f001:**
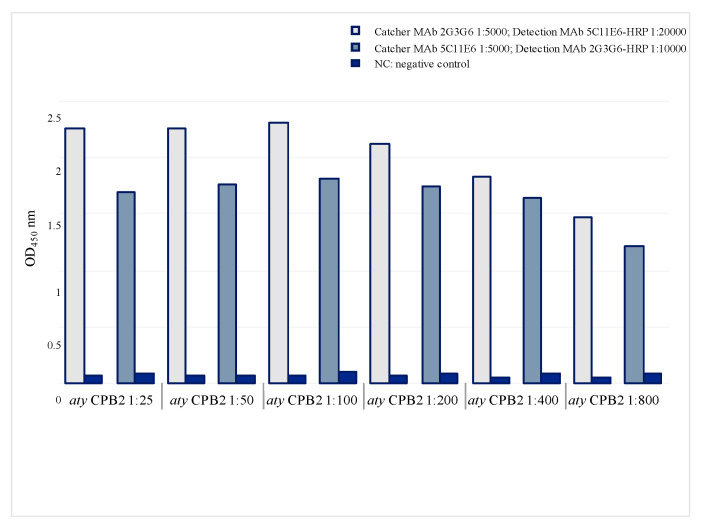
Capture abilities of MAbs 2G3G6 and 5C11E6 in sELISA. Dilutions of *C. perfringens* atypical *cpb2*+ strain culture supernatant (*aty* CPB2) (*x*-axis) vs. OD_450_ values (*y*-axis). NC: negative control (*C. perfringens cpb2*- strain culture supernatant).

**Figure 2 toxins-14-00796-f002:**
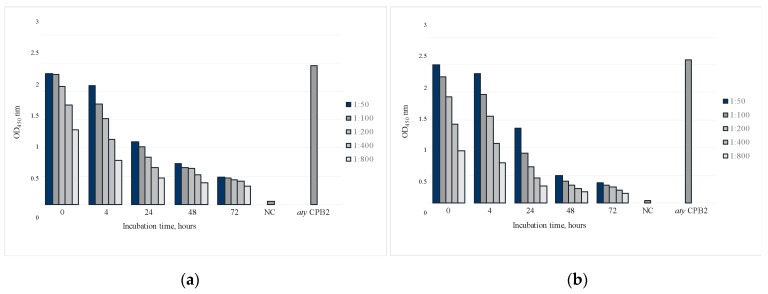
sELISA of artificially contaminated faeces with *C. perfringens* atypical *cpb2*+ strain culture supernatant (*aty* CPB2). (**a**) sELISA with MAb 5C11E6-HRP as detection system; (**b**) sELISA with MAb 2G3G6-HRP as detection system. Incubation time (*x*-axis) at room temperature, *aty* CPB2 dilutions vs. OD_450_ values (*y*-axis). NC: negative control (*C. perfringens cpb2*- strain culture supernatant).

**Figure 3 toxins-14-00796-f003:**
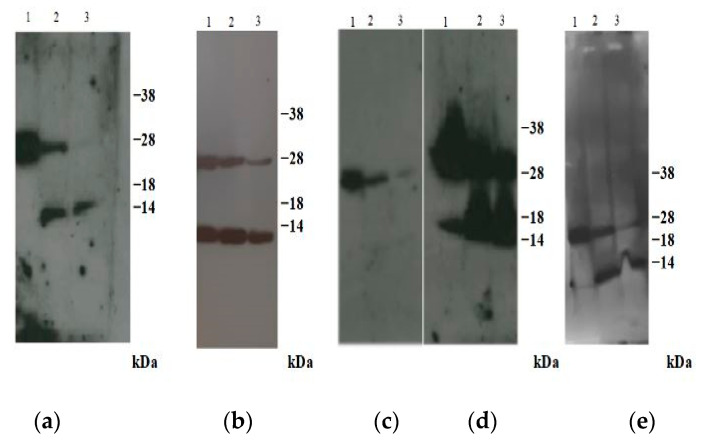
Western blotting of artificially contaminated feces with *C. perfringens* atypical *cpb2*+ strain culture supernatant. (**a**) Hyperimmune serum and anti-rabbit HRP-conjugated MAb as detection system; (**b**) MAb 2G3G6; (**c**) MAb 5B4G9G1; (**d**) MAb 3G4C8; (**e**) MAb 2G3D10. (**b**–**e**): anti-mouse HRP-conjugated MAb as detection system. Incubation time t=o (*lane 1*); incubation time t = 4 h (*lane 2*); incubation time t = 24 h (*lane 3*). Standard molecular weight is shown on the right.

**Figure 4 toxins-14-00796-f004:**
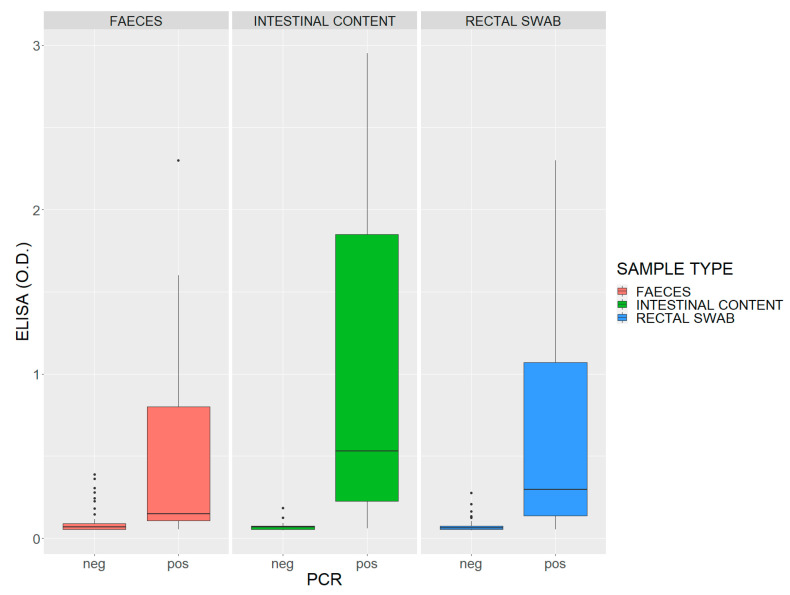
Box plot of quantitative sELISA distribution vs. PCR qualitative response in each different sample type. OD_450_ values (*y*-axis) and PCR outcomes (*x*-axis) are shown.

**Table 1 toxins-14-00796-t001:** Reactivity variations among 5C11E6, 2G3G6, 4E10E11 and conformation-dependent 23E6E6 MAbs in sELISA ^1^.

Strain Number	MAbs			
	5C11E6	2G3G6	4E10E11	23E6E6
C1	2.4	2.4	2.66	0.38
C2	1.83	1.32	0.62	0.9
C3	1.71	0.75	0.68	0.4
C4	0.571	0.225	0.27	0.145
C5	0.54	0.21	0.207	0.082
C6	2.65	2.85	2.67	0.17
C7	1.29	0.49	0.44	0.153
C8	2.63	0.82	0.29	2.11
C9	0.47	0.33	0.226	0.163
C10	2.69	2.44	2.3	0.44
C11	0.56	0.298	0.219	0.088
C12	1.5	0.38	0.45	0.29
C13	0.4	0.204	0.17	0.084
C14	2.3	2.35	2.42	0.2
C15	0.576	0.578	0.372	0.106
C16	2.5	2.45	2.46	0.12
C17	1.63	0.71	0.72	0.58
C18	0.239	0.16	0.093	0.111
C19	0.1	0.145	0.6	0.072

^1^ OD_450_ values are shown.

**Table 2 toxins-14-00796-t002:** Detection of atypical CPB2 on field samples with both PCR and sELISA ^1^.

Sample Number	Faeces		Intestinal Content		Rectal Swab	
	sELISA	PCR	sELISA	PCR	sELISA	PCR
31	0.946	Positive	1.87	Positive	0.21	Positive
32	2.3	Positive	2.83	Positive	1.08	Positive
33	0.78	Positive	2.4	Positive	1.42	Positive
34	0.75	Positive	0.14	Positive	1.3	Positive
35	0.112	Positive	1.15	Positive	0.585	Positive
36	0.12	Positive	0.492	Positive	0.304	Positive
37	0.054	Positive	1.37	Positive	0.29	Positive
38	0.83	Positive	0.079	Positive	2.27	Positive
39	0.105	Positive	2.48	Positive	0.081	Positive
40	0.068	Positive	1.18	Positive	0.054	Positive
41	0.113	Positive	0.059	Positive	0.069	Positive
42	0.088	Positive	0.426	Positive	0.117	Positive
43	ND	ND	0.5	Positive	0.55	Positive
44	ND	ND	2.85	Positive	2.3	Positive
45	ND	ND	2.95	Positive	0.195	Positive
46	ND	ND	0.117	Positive	0.1	Positive
47	1.6	Positive	0.43	Positive	ND	ND
48	0.161	Positive	2.2	Positive	ND	ND
49	1.24	Positive	ND	ND	ND	ND
50	ND	ND	1.28	Positive	ND	ND
51	0.133	Positive	0.1	Negative	0.098	Positive
52	0.348	Positive	0.375	Positive	0.066	Negative
53	0.05	Negative	0.43	Positive	0.217	Positive
54	0.15	Positive	0.132	Positive	0.064	Negative
55	0.072	Positive	2	Positive	0.055	Negative
56	0.062	Positive	2.16	Positive	0.049	Negative
57	0.104	Negative	0.099	Positive	0.088	Negative
58	0.065	Negative	2.25	Positive	0.128	Negative
59	0.362	Negative	1.77	Positive	0.135	Negative
60	0.069	Negative	1.4	Positive	0.06	Negative
61	0.065	Negative	0.75	Positive	0.093	Negative
62	0.057	Negative	0.129	Positive	0.053	Negative
63	0.057	Negative	0.24	Positive	0.052	Negative
64	0.062	Negative	0.153	Positive	0.063	Negative
65	0.076	Negative	0.7	Positive	0.063	Negative
66	0.053	Negative	0.56	Positive	0.05	Negative
67	0.051	Negative	0.23	Positive	0.048	Negative
68	0.06	Negative	0.11	Positive	0.066	Negative
69	0.145	Negative	0.306	Positive	0.277	Negative
70	0.225	Negative	0.22	Positive	0.055	Negative
71	0.087	ND	2.25	Positive	0.063	Negative
72	ND	ND	0.471	Positive	0.105	Negative
73	0.244	Uncertain	0.048	Negative	0.101	Negative
74	0.18	Uncertain	0.068	Negative	0.075	Negative

^1^ OD_450_ values and PCR outcomes are shown. Intensity of PCR bands is depicted by greyscale (see also Appendix A). ND: not determined.

**Table 3 toxins-14-00796-t003:** Friedman test.

	Faeces	Intestinal Content	Rectal Swab
Faeces	1		
Intestinal content	0.043	1	
Rectal swab	0.061	<0.0001	1

## Data Availability

Not applicable.

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
