# Peer review of "Identification and Characterization of *Clostridium perfringens* Atypical CPB2 Toxin in Cell Cultures and Field Samples Using Monoclonal Antibodies"

_toxins, 2022, doi:10.3390/toxins14110796_

Round 1

Reviewer 1 Report

Clostridium perfringens is anaerobic Gram-positive microorganism, causing mild to severe
enteric diseases in human population and animals. Major virulence factors of the bacterium
include toxins, which, in addition to participation in the pathogenesis, represent a basis for

C. perfringens typing. Therefore, correct detection of the toxins in environmental or clinical
samples is very important. The reviewed paper is dedicated to characterization of monoclonal
antibodies, able to detect atypical forms of beta-toxin2. The topic in interesting, experiments
are well-planned and the results are clearly-presented.
However, from my point of view, several issues should be addressed before publishing the
manuscript.
Majors.
1. There are quite a lot of language problems. Native English speaker should check style
and grammar.
2. Title. I would delete “and Polyclonal” from the title. PAbs were not deeply investigated
in the study and have been used only as a “non-specific” catcher in the assay. Their
usage was necessary but not scientifically sound
per se. Between the lines, I would say
that theoretically, PAbs can be somewhat “specific” and can catch some forms of CPB2
more efficiently, thus making a bias in the results with subsequently used MAbs. But this
is my speculation, forget it.
3. Results. There is an obvious confusion with numerical data (see below Minors).
4. Discussion. This section is very redundant. Many parts of it are just a simple
reproduction of the Results section, rather than analysis of the obtained results and
their relationship with literature data. The section should be shortened significantly.
Minors.
5. Line 8. “In this method…”. Better to say e.g. “In the current study…”
6. Line 10. Why "
wild type” is accented by italics? The meaning is unclear. Delete?
7. Line 38-39. Language problems. “In spite of…” ?
8. Line 43. “Benz et al…”
9. Lines 46-49. Language problems.
10. Line 57-58. EXPRESSION of the gene, while PRODUCTION (or SYNTHESIS) of the toxin
11. Line 79. Again “
wild type”. Not clear for me.
12. Line 79. Worth saying that it is a crude protein (liquid culture sample), not purified.
Should be indicated also in the Fig1 and throughout in the text.
13. Line 80. 5+2≠6
14. Figure 1. Here and in other figures “Negative controls” are shown. But what represent
NCs? I did not find explanation.
15. Line 98 and further. “…using atypical CPB2 produced by
C. perfringens strains.” Not
clear. Does it mean that only 36 atypical strains have been tested. At the same time, you
say “Specifically, 36 atypical
cpb2+ strains, 5 consensus cpb2+ and 2 cpb2- strains were
analysed by sELISA using the whole MAbs panel.” Further, “The results obtained by
sELISA showed high OD450 values for the atypical
cpb2+ strains (36 strains?) with all
tested MAbs…”. What about 5 consensus and 2 negatives?
16. Line 104,105. “Linear…Conformational…”. Time and again you tell this. Please give more
info.

17. Line 107. “79 samples” Why 79?
18. Line 109. “In other 19 samples…” Other 19? In general, numbers are confusing for me,
do not fit to each other and difficult to understand
19. Line 117. “The remaining 7 atypical strains…”. Uff… I am totally confused.
20. Line 136. Please, indicate temperature of incubation (RT?). It is important.
21. Line 145. “Figure 3”. Should be “Figure 3a”. “…entire protein” should be “full size
protein”.
22. Line 149. Data with 5B4G9G1 (Fig. 3c) are questionable. 10-12kD band is not clearly
seen.
23. Line 157. “…BY biomolecular method”. “Biomolecular method” is PCR? Not very precise
term.
24. Table 2. Not very informative table. Maybe a description in the textbody would be
sufficient? The table can go to Supplementary.
25. Line 165. Why did not you mention samples 26, 27 (feces) and 25, 27 and 29 (swab)?
26. Table 3. What is N/P (strains 73 and 74)?
27. Line 171. How did you scale “intensity of PCR bands”? Maybe representative figures
could be put into Supplementary?
28. Lines 184-187. Duplication with Lines 163-166 ?
29. Lines 192-193. Language problems, not clear.
30. Figure 4. Should be improved. Difficult-to-understand figure.
31. Lines 201-206. Language problems
32. Line 210. “…several animal and human diseases…”.
33. Line 236. “Surprisingly…” It is not surprising - expression level, mutations etc. Many
influences.
34. Line 291. “…the increase in noise…” What does it mean?
35. Line 321-324. Unclear sentence. Please re-phrase.
36. Line 344. “…an improvement of the sELISA method…” Your data indicate toward
improvement of sample processing, not ELISA method
per se.
37. Line 360-361. “The results obtained on BACTERIAL cell culture…”
38. Line 368. No info on production of PAbs, however you speak a lot about them.
39. Line 411, 417 etc. What is “1% yeast” ?

Reviewer 2 Report

Dear Authors,

The topic of the manuscript entitled ‘Identification and characterization of Clostridium perfringens Atypical CPB2 toxin in cell cultures and field samples using Monoclonal and Polyclonal antibodies’ is interesting but needs some changes.  

comments:
Line 22 – instead of ‘C. perfringens ‘ it should be ‘Clostridium perfringens’ and instead of CPB2 toxin it should be atypical β2-toxin

Lines 210-211 – this statement has already been used in the introduction. Please deleted it from Discussion section.

Poor resolution of Figure 1.

Round 2

Reviewer 1 Report

All my comments have been adequatelly addressed.

Reviewer 2 Report

I accept in a present form